# 12-Week Exercise Training of Knee Joint and Squat Movement Improves Gait Ability in Older Women

**DOI:** 10.3390/ijerph18041515

**Published:** 2021-02-05

**Authors:** Myungsoo Choi, Nayoung Ahn, Jusik Park, Kijin Kim

**Affiliations:** Department of Physical Education, College of Physical Education, Keimyung Univesity, 1095 Dalgubeul-daero, Dalseo-gu, Daegu 42601, Korea; halfno9@gmai.com (M.C.); nyahn13@kmu.ac.kr (N.A.); parkjs@kmu.ac.kr (J.P.)

**Keywords:** gait, exercise, aging, physical fitness, fall

## Abstract

This study analyzed the effects of an exercise training program consisting of a knee joint complex exercise device (leg-link system) with digitally controlled active motion function and squat movement on physical fitness and gait ability of elderly women aged 70 or above. Fifty four (54) elderly women aged 70 or above were divided into three groups as control group (*n* = 18), aerobic training group (*n* = 18), and combined training group with resistance and aerobic exercise (*n* = 18). Health-related physical fitness, gait ability-related physical fitness, and the temporal and spatial parameters of gait ability were compared. The health-related physical fitness after the 12-week training was not significantly altered in control group, whereas combined training group showed significant increase in all factors (*p* < 0.05) and aerobic training group showed significant increase (*p* < 0.05) only in the physical efficiency index. The gait ability-related physical fitness and all items of the temporal and spatial parameters of gait were found to have significantly increased (*p* < 0.05) in combined training group after the 12-week exercise training; however, in aerobic training group, only the factors related to muscular endurance and balance showed significant increase (*p* < 0.05). This study suggested that the exercise training consisting of knee joint complex exercise with digitally controlled active motion function and squat exercise for strengthening lower extremities and core muscles had positive effects on enhancing the ambulatory competence in elderly women.

## 1. Introduction

Recently, the retention of ambulatory competence based on preventing muscle mass reduction and falls has received growing attention as an important factor in health maintenance during aging. Ambulatory competence is determined by physical, psychological, and habitual factors, and it is considered a crucial index of aging that is inextricably linked to a wide range of physical functions including cognitive functions [1]. The physical fitness factors that affect ambulatory competence are the lower limb and core muscle strengths and balance so that the decrease in muscle strength due to muscle mass reduction in aging is a critical factor that impairs gait ability [2].

In particular, decrease in the lower limb and core muscles leads to their functional impairment and causes problems in gait ability, while inducing fall accidents of the elderly with direct influence on their mortality [3]. Thus, to prevent reduced gait ability and fall accidents, a method of increasing the muscle mass in the core and lower extremities and strengthening muscular function is required, and a relevant intensive training is necessary [4]. Most importantly, an effective resistance exercise training program based on resistance exercise for adequately loading the lower limb and core muscles should be carried out for increasing the muscles in lower extremities and improving gait ability [5].

The gait of the elderly is characterized by the decrease in walking velocity and stride width as well as the increase in movement instability [6], and the reduced overall body function such as reduced grip strength leads to frailty, a factor that has been reported to exhibit strong association with reduced ambulatory competence [7]. Furthermore, reduced function in the lower extremities has crucial influence on falls, a factor regarded as one of the main causes of death [8]. In this respect, the importance of improving physical fitness to prevent fall accidents should be emphasized. 

The reduced ability to perform activities in aging has a substantial impact on the decrease in quality of life, and as a basic capability related to this, the ability to walk, run, and lift objects was suggested and with it came the increased attention on the importance of relevant physical fitness level [9]. For maintaining or improving gait ability, it seems essential that an exercise training program be designed for improving power of whole body while focusing on increasing the muscle mass in lower extremities and enhancing the muscular function. The need for exercise programs and equipment that can be used to promote active aging and health of the elderly while aiming for a safe and physical fitness level is being emphasized [10]. In particular, the importance of the development of an exercise program that can show a higher clinically positive effect in the field safely in order to improve the gait ability of elderly women is more emphasized. Therefore, this study aims to analyze the changes between the level of improvement in muscle function and factors related to gait ability, based on the recognition that the gait dysfunction caused by sarcopenia during aging is mainly due to the decrease in the muscle function of the core and lower extremities. 

Notably, by providing a personal exercise training program with knee joint complex exercise device (leg-link system) with digitally controlled active motion function and squat exercise developed from a complementary viewpoint for activation of lower limb and core muscles [11] to elderly women aged 70 or above, and the study aims to analyze the efficacy of the exercise training program. 

## 2. Materials and Methods

### 2.1. Study Design

In order to analyze the exercise training effect of the subjects, this study measured the items such as body composition, health-related physical fitness, gait ability-related physical fitness, and ambulatory competence, respectively, before and after 12-week exercise training, and compared the change patterns. Body composition and physical fitness were measured while fasting for at least 8 h was maintained. The participants were divided into control group, aerobic training group, and combined training group. The subjects in aerobic training group and combined training group were asked to participate in the exercise training program for at least three times a week. Control group performed only general life while being controlled not to perform exercise program for 12 weeks. We obtained approval for the recruitment and exercise treatment experiment from the Institutional Review Board (IRB) of Keimyung University (40525-201802-BR-1218-02).

### 2.2. Participants

In order to calculate the appropriate subjects for three groups to secure statistical significance, the G-power 3.1.9.2 program (HHU, Dusseldorf, NRW, Germany) was used to analyze based on the effect size of 0.5 and the power of 0.95. As a result, the total sample size was calculated as 51 subjects. In consideration of the dropout rate of about 20% or more, a total of 62 elderly women aged 70 or above who voluntarily participated in the K University health management program. Those who dropped out due to personal reasons, and those whose test results were not reliable were excluded. The participants were divided into control group, aerobic training group, and combined training group, each group containing 18 subjects. Among the subjects of this study, cases dependent on drug use were excluded, and subjects without health problems through a medical examination by a doctor were composed.

### 2.3. Exercise Training Program

The aerobic training consisted only of aerobic exercise including walking and light jogging. The exercise intensity was 60–70% maximum heart rate and exercise time was 50 min. The combined training consisted of resistance exercise and aerobic exercise including walking and light jogging. The intensity per exercise type was 60–70% maximum heart rate for aerobic exercise and 60–70% of 1RM for resistance exercise based on 3 sets of 15RM. The resistance exercise training program focused on knee joint complex exercise device (leg-link system) with digitally controlled active motion function and squat exercise developed from a complementary viewpoint for activation of lower limb and core muscles [11], and to examine which muscles are used, electromyographic analysis was performed. For this, two surface electrodes of 10 mm diameter were attached to the leg muscles including vastus medialis, hamstrings, gastrocnemius, tibialis anterior, and soleus, in 25 mm interval for the estimation using the electromyography system (WEMG 8, Laxtha Inc., Seoul, Korea). The exercise time was based on 12-week personal exercise training program and exercise intensity was redesigned in four weeks interval. The total exercise time was 60 min including the warming-up and cool-down exercises. The combined training program was constructed according to FITT Recommendations for the elderly set by the American College of Sports Medicine [12] as shown in Table 1.

### 2.4. Body Composition

The body weight was measured using InBody 3.0 (Biospace, Seoul, Korea) and the body mass index (BMI) was calculated according to the equation ‘body weight (kg)/height (m^2^)’. For body fat, the subcutaneous fat thickness in subscapular, abdomen, and iliac crest were measured using Skinfold caliper (Skyndex, Fabrication Enterprises, New York, NY, USA) which was used to calculate the body density according to equation [13], the value being translated to % using %fat equation [14]. For body circumference, a tape measure was used to estimate the waist and hip circumferences: For waist circumference, the WHO [15] criteria was used so that it was measured at the middle area between the lower part of rib 12 and the upper part of iliac crest, and based on this, the waist to hip ratio (WHR) was calculated.

### 2.5. Health-Related and Gait Ability-Related Physical Fitness

The health-related physical fitness was estimated based on grip strength, back muscular strength, sit-up, sit & reach, and Harvard step test. To measure grip strength and back muscular strength, the grip strength meter (TKK, Tokyo, Japan) and the back muscular strength meter (TKK), respectively, were used. The sit & reach was recorded during two-second static state of the subject whose upper body was completely bent with arms reaching out as far as possible while sitting on the measuring device with knees straightened out and both soles completely touching the vertical surface of the device. The sit-up was recorded as the number of repeated movements during 30 s, where a movement was defined as the subject lying on a mat with knees bent to approximately 140° and soles touching the ground while the upper body was raised so that the elbows touched the knees. For the Harvard step test, the subjects performed going up and down a 35 cm high ramp 30 times a minute for 3 min. When all exercises were complete, the pulse of the subject was measured between 1 min and 1 min 30 s; 2 min and 2 min 30 s; 3 min and 3 min 30 s, and the equation below was used to convert the measured values into physical efficiency index (PEI) as equation ‘PEI = [Exercise time (s)/Sum of 3 times measured pulse rate × 2] × 100’.

The gait ability-related physical fitness was estimated based on S-style walking, temporal-distance gait, sit-up, standing on one leg with eyes open, Time Up Go, and straight-distance gait, following the methods in Moniz-Pereira et al. [16]. For straight-distance gait, a stopwatch was used to record the time taken for the subject to walk 8 m distance from a starting point to a finishing point. For S-style walking, the time taken for the subject to walk 3 m lane shaped as the number 8 from a starting point to a finishing point was recorded. For Time Up Go, the time taken for the subject to start at a sitting position from which he or she stood up with the start sound, walked briskly on 3 m lane, then returned to the initial sitting position was measured. For standing on one leg and sit-up, the time taken for the subject to perform the activity within 1 min was recorded. The straight-distance gait was measured as the time taken for the subject to walk 3 m distance between a starting point and a finishing point, while he or she was asked to place the toe of the back foot on the heel of the front foot then repeating the movement during the entire walk.

### 2.6. Ambulatory Competence

To analyze the temporal and spatial parameters of gait, GAITRite (MAP/CIR, Inc. Franklin, TN, USA) was used [17]. The measured data was categorized into temporal gait parameters (step time and mean velocity) and spatial gait parameters (step length, stride length, step to extremity ratio, and heel-heel (H-H) base of support). The step time in temporal parameters refers to the time taken between the contact of one leg and that of the opposite leg (i.e., from the heel of right foot to the heel of left foot). The gait cycle refers to the time taken between the initial contact of the moving leg and the next contact (i.e., from the heel of right foot at one time to the heel of right foot at the next). The single support time refers to the time of contact between one leg and ground surface, and double support time refers to the time of simultaneous contact between both legs and ground surface. The ambulation time refers to the time between the first step and the last step, and mean normalized velocity refers to the mean length of the two legs divided by velocity. In spatial parameters, the stride width refers to the perpendicular distance between the line of progression and heel location of the contralateral foot (i.e., from the heel of right foot to the heel of left foot) while stride length refers to the distance between heel locations of two consecutive footfalls of the same foot. The step to extremity ratio refers to the step length divided by the length of the moving leg. The H-H base of support refers to the horizontal line between the heel of one foot and the forward line of the opposite foot.

### 2.7. Statistical Analysis

For data analysis, SPSS 23.0 for Window statistical program (IBM, New York, NY, USA) was used for producing the mean and standard deviation for the measured data of all factors. To analyze the difference between the time before exercise training program and the time after the training, 2-way repeated ANOVA was carried out with the group and period of time as independent variables after first testing the homogeneity of the measurement results. As post-hoc, paired t-test was carried out for time per group and one-way ANOVA test with Tukey test was carried out for group per time. The level of significance was set as *p* < 0.05.

## 3. Results

Enrollment flow chart of subjects is shown in Figure 1. The physical characteristics of the subjects are shown in Table 2.

The changes in body composition after the 12-week exercise training are shown in Figure 2. In contrast to control group that maintained almost identical composition before and after the training, aerobic training group and combined training group showed significantly decreased (*p* < 0.05) body weight, body fat, and BMI after the training, and lean body mass, in particular, showed significant increase (*p* < 0.05) only in combined training group.

The changes in health-related physical fitness after the 12-week exercise training are shown in Figure 3. Combined training group showed significant increase in all areas including grip strength, back muscular strength, sit-up, sit & reach, and physical efficiency index (*p* < 0.05) after the exercise training, whereas Control group did not show any significant change and Aerobic training group showed significant increase (*p* < 0.05) only in the physical efficiency index, a factor related to cardiopulmonary function.

The changes in gait ability-related physical fitness after the 12-week exercise training are shown in Figure 4. Combined training group showed significant increase in S-style walking velocity, 8 m walking velocity, and Straight walking velocity that are related to the walking velocity corresponding to gait ability (*p* < 0.05) after the training. Combined training group also showed significant increase in up & down frequency in chair indicating muscular endurance and in one-leg balance indicating balance capacity and in Time Up Go indicating agility (*p* < 0.05) after the training. In contrast, aerobic training group showed significant increase only in up & down frequency in chair indicating muscular endurance and in one-leg balance indicating balance capacity (*p* < 0.05) after the training. Control group did not show any significant change in all areas before and after the training.

The temporal and spatial parameters of gait are shown in Table 3 and Table 4, respectively. In contrast to combined training group exhibiting significant increase (*p* < 0.05) in all items after the 12-week exercise training, aerobic training group and control group did not show significant differences in all items before and after the exercise training.

## 4. Discussion

In this study, after the 12-week exercise training, combined training group showed positive changes in body composition accompanied by improvement in the physical fitness factors, the individual parameters that influence the ambulatory competence, and parameters related to gait ability, especially gait pattern. Thus, the resistance exercise training focusing on knee joint complex exercise device (leg-link system) with digitally controlled active motion function and squat movements for strengthening the lower limb and core muscles, respectively, seems to have had positive influence on physical fitness factors that are crucial in ambulatory competence and on temporal and spatial parameters of gait ability, thereby exerting outstanding effects on improving the ambulatory competence.

Ambulatory competence is a key indicator of the overall physical functions of the body as it corresponds to the most fundamental ability underlying independent activities of the elderly as well as the prolonged lifespan and healthy lifestyle [18]. The most general phenomenon of aging is reduced ambulatory competence caused by decreased walking velocity and step frequency, increased stride width and double support time, and decreased step length [19]. Of particular importance are reduced stride width and stride length that negatively affect the walking velocity, a key parameter for ambulatory competence [20].

The most basic factor that brings about reduced ambulatory competence is probably the reduction in physical fitness. The central factors of reduced step length and stride length that influence the walking velocity are considered to be related to the reduced physical fitness around lower extremities such as muscular strength during ankle movement, flexibility, and strength of flexor and extensor muscles at the knee joint [21]. Furthermore, reduced physical fitness has been regarded as a critical risk factor in fall accidents as it decreases the stride width and stride length that are related to ambulatory competence of the elderly [22].

Notably, the increased walking velocity after the exercise training in this study is thought to have the potential to substantially influence the prevention of fall accidents [23]. Regarding this, Toulotte et al. [24] reported that, when the subjects with experience of a fall accident were given an exercise training focusing on improving the muscular strength, flexibility, and balance, their gait ability was enhanced and the probability of falls markedly decreased. In addition, according to Ramalho et al. [10], although a program for enhancing the ability to perform activities to improve the general ability to maintain posture, balance, and lower limb muscular strength for aerobic exercise, definitely helps to enhance the ambulatory competence in the elderly based on improved physical fitness, further research is required to prove the effects of physical fitness improvement on individual gait patterns. However, as physical fitness and the individual parameters of gait were improved simultaneously in this study, the improved physical fitness is likely to enhance the overall gait pattern and walking velocity. In this study, aerobic training group was shown to have had positive influence in terms of the physical fitness factors related to endurance but not significantly in terms of the individual parameters of gait. On the contrary, combined training group where resistance training was combined with aerobic training showed marked improvement in both the physical fitness factors and the individual parameters of gait. Thus, to improve the individual parameters related to gait ability, not only the overall physical fitness improvement but also the efforts to improve the muscular functions at the core and lower extremities seem essential.

The individual parameters of gait ability is known to be under significant influence of the sensorimotor system [25]; nevertheless, of utmost importance is basic physical fitness factors, especially the extensive influence from the muscular functions at the lower extremities and the strength, flexibility, and balance from the core muscles. Comparing the gait characteristics in the elderly with those in youth shows a tendency for walking velocity to decrease and step length to shorten. Also, for climbing slopes or increasing velocity, the reduced joint mobility may cause decrease in the eversion angle inside the hip or knee joint, which consequently increases the rotation angle of the leg upon walking and the rotation time, thereby increasing the instability towards ground surface and leading to a fall [6]. The gastrocnemius and soleus muscles involved in ankle joint motion are especially important in gait instability [26], and such instability due to reduced muscular function may appear in patients with brain damage after concussion [27].

To complement this, a program is required for improving muscular strength via the movement of the entire lower extremities and for strengthening the ankle and knee joints as well as enhancing the core muscular function. Thus, knee joint complex exercise with digitally controlled active motion function and squat movements for strengthening the complex movements of the hip, knee, and ankle joints and for strengthening the core muscles, respectively, seem to have provided direct assistance to the enhancement of gait in the elderly. As shown in Figure 5, analyzing the muscles recruited for the knee joint complex exercise with digitally controlled active motion function and squat movements revealed that the muscles involved in ankle joint motion were most active.

The negative changes in factors related to decreased swing phase are thought to have crucial impact on reduced ambulatory competence in the elderly [28]. According to Iwamoto et al. [29] where the individual parameters of gait were analyzed after requesting elderly women aged 70 or above to perform the 12-week training combining resistance exercise and balance exercise, all temporal and spatial parameters were shown to have improved; especially, the resistance exercise for enhancing the lower limb muscular functions was shown to have had positive effects on the parameters related to gait ability. Exercise training can help enhance the ambulatory competence by improving step cadence and stride length [30] and a study reported the positive effects of a combined training for improving muscular functions and cardiopulmonary functions to enhance the ambulatory competence in the elderly [31]. In particular, Ramalho et al. [10] claimed the ambulatory competence in the elderly could be enhanced based on improved physical fitness in regard to functional activities after an effective exercise training. This opinion support to the report in this study that the knee joint complex exercise with digitally controlled active motion function and squat movements should be included in the training program with outstanding effects on improving gait ability-related physical fitness and the individual parameters related to gait ability in the elderly.

Ramalho et al. [10] emphasized the improvement in postural control by reduced plantar pressure, based on the fact that physical fitness improvement from enhanced lower limb muscular strength via exercise training is a critical factor in bringing about positive changes to the gait pattern in the elderly. Notably, such factors were also emphasized in Burton et al. [32] who pointed out the increased plantar pressure in the hindfoot zone upon heel contact is a crucial factor reducing the gait pattern stability and inducing falls in the elderly [10]. This study showed increased stride width and step length among the changes in the individual parameters of gait after the exercise training, and muscular functions involved in ankle joint motion play an important role in increasing such factors [21]. Thus, it is probable that the movements using the knee joint complex exercise device (leg-link system) with digitally controlled active motion function employed in this study had been effective for improving the ankle joint motion. Furthermore, the group that performed the exercise training focusing on knee joint complex exercise device (leg-link system) with digitally controlled active motion function movements and strengthening core muscles exhibited positive improvement in the individual spatial and temporal parameters of gait, while most of the physical fitness factors that influence walking velocity as well as the actual walking velocity were also positively improved.

It is considered to be a significant influencing factor in reducing stride in older adults, such as decreased arm swing, decreased range of motion in the hips, knees and ankles, increased double support time, and HH bass [26]. Most of these factors were found to be improved in the group that performed exercise training focused on the digitally controlled active motor function knee joint complex exercise and squat exercise investigated in this study. Such factors were clearly shown to have enhanced the stride width and gait velocity.

The tendency to fall more often during walking is associated with a number of factors [33]. Especially, exercise intervention to improve muscle strength reduce the rate falls by 23%, but resistance training alone may not lead to reductions in falls [34]. Active control of upright balance during walking requires a sensorimotor control loop that collects sensory information about the movement of the body space, detects deviations from the upright posture, and generates appropriate muscle forces to correct these deviations [35]. Therefore, exercise training consisting of knee joint complex exercise with digitally controlled active motion function and squat exercise is thought to be helpful in improving sensorimotor function along with lower limb muscle function in the elderly. In the case of the elderly, the importance of rehabilitation training to improve walking ability is emphasized, so if a feedback function appropriate to the level of muscle strength is used, the possibility of using it in the actual clinical field will increase.

From the perspective of a pilot study [36], the research team analyzed the training effects including knee flexion and squat movements for elderly women. The degree of improvement in cognitive function showed a significant correlation coefficient (*r* = −0.345, −0.341; *p* < 0.05) with the degree of improvement in linear walking ability and Time Up Go performance. Therefore, even after the treatment of exercise training applied in this study, it is expected that the possibility of application of the applied training method can be further enhanced by more accurately analyzing how the improvement ranges such as body composition, physical fitness level, and ambulatory competence represent each other.

## 5. Conclusions

This study investigated the effects of the exercise training consisting of the knee joint complex exercise with digitally controlled active motion function and squat movements for strengthening the lower limb and core muscles, and suggested that it could enhance the ambulatory competence in elderly women aged 70 or above by improving their gait ability-related physical fitness and the individual parameters of gait.

## Figures and Tables

**Figure 1 ijerph-18-01515-f001:**
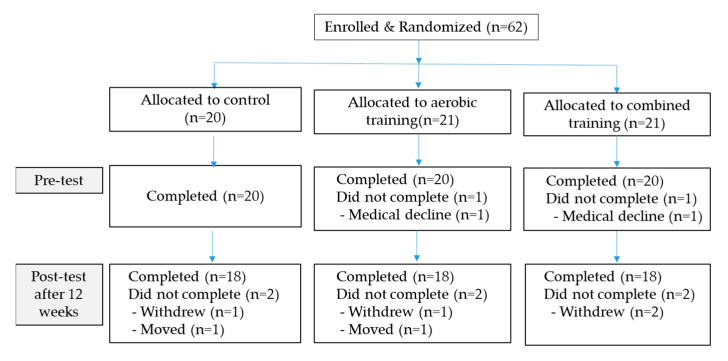
CONSORT diagram of subjects.

**Figure 2 ijerph-18-01515-f002:**
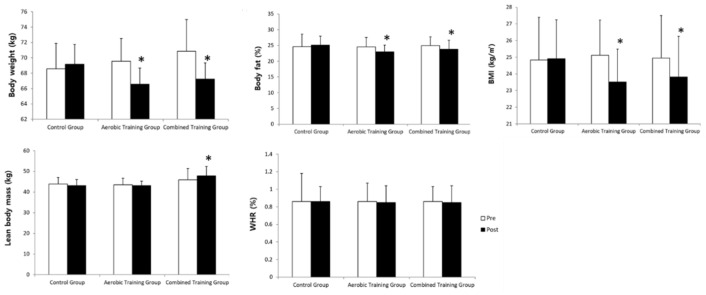
Change of body composition after 12 week exercise training. 2-way repeated ANOVA for group and time: Body weight, Group *F*_2,51_ = 3.234 (*p* < 0.05), Time *F*_1,51_ = 5.345 (*p* < 0.05), G × T *F*_2,51_ = 14.512 (*p* < 0.05); Body fat, Group *F*_2,51_ = 4.167 (*p* < 0.05), Time *F*_1,51_ = 3.317 (*p* < 0.05), G × T *F*_2,51_ = 4.512 (*p* < 0.05); BMI, Group *F*_2,51_ = 4.177 (*p* < 0.05), Time *F*_1,51_ = 4.117 (*p* < 0.05), G × T *F*_2,51_ = 8.912 (*p* < 0.05); Lean body mass, Group *F*_2,51_ = 4.231 (*p* < 0.05), Time *F*_1,51_ = 2.356 (*p* > 0.05), G × T *F*_2,51_ = 5.534 (*p* < 0.05); WHR, Group *F*_2,51_ = 2.189 (*p* > 0.05), Time *F*_1,51_ = 1.389 (*p* > 0.05), G × T *F*_2,51_ = 2.667 (*p* > 0.05). *: *p* < 0.05 vs. Pre-test.

**Figure 3 ijerph-18-01515-f003:**
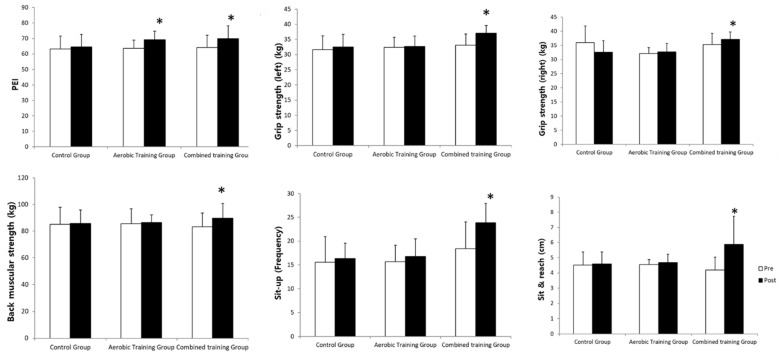
Change of health-related physical fitness after 12 week exercise training. 2-way repeated ANOVA for group and time: PEI, Group *F*_2,51_ = 4.131 (*p* < 0.05), Time *F*_1,51_ = 5.129 (*p* < 0.05), G × T *F*_2,51_ = 9.167 (*p* < 0.05); Grip strength (Left), Group *F*_2,51_ = 4.123 (*p* < 0.05), Time *F*_1,51_ = 5.661 (*p* < 0.05), G × T *F*_2,51_ = 5.178 (*p* < 0.05); Grip strength (Right), Group *F*_2,51_ = 3.987 (*p* < 0.05), Time *F*_1,51_ = 4.890 (*p* < 0.05), G × T *F*_2,51_ = 6.617 (*p* < 0.05); Back muscular strength, Group *F*_2,51_ = 4.002 (*p* < 0.05), Time *F*_1,51_ = 4.919 (*p* > 0.05), G × T *F*_2,51_ = 9.617 (*p* < 0.05); Sit-up, Group *F*_2,51_ = 5.977 (*p* < 0.05), Time *F*_1,51_ = 4.657 (*p* < 0.05), G × T *F*_2,51_ = 9.111 (*p* < 0.05); Sit & reach, Group *F*_2,51_ = 6.134 (*p* < 0.05), Time *F*_1,51_ = 5.919 (*p* < 0.05), G × T *F*_2,51_ = 8.268 (*p* < 0.05). *: *p* < 0.05 vs. Pre-test. *: *p* < 0.05 vs. Pre-test.

**Figure 4 ijerph-18-01515-f004:**
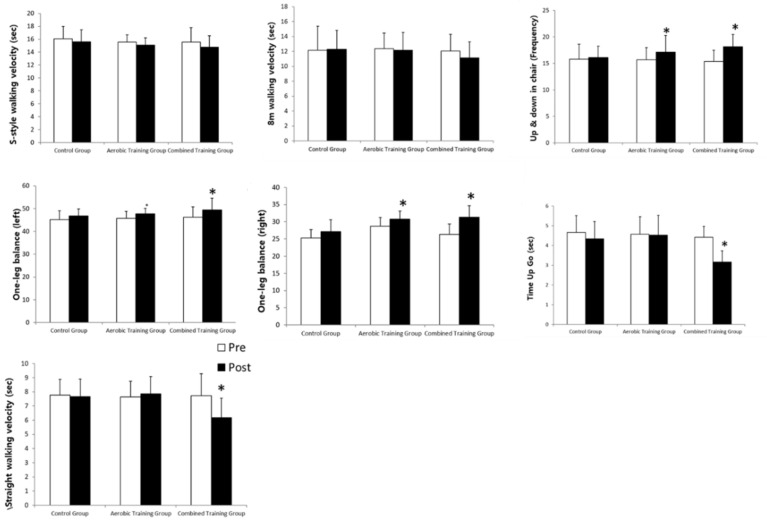
Change of gait ability-related physical fitness after 12 week exercise training. 2-way repeated ANOVA for group and time: S-style walking velocity, Group *F*_2,51_ = 2.131 (*p* > 0.05), Time *F*_1,51_ = 4.671 (*p* < 0.05), G × T *F*_2,51_ = 8.357 (*p* < 0.05); 8 m walking velocity, Group *F*_2,51_ = 1.911 (*p* > 0.05), Time *F*_1,51_ = 2.112 (*p* > 0.05), G × T *F*_2,51_ = 1.196 (*p* > 0.05); Up & down in chair, Group *F*_2,51_ = 5.551 (*p* < 0.05), Time *F*_1,51_ = 5.981 (*p* < 0.05), G × T *F*_2,51_ = 9.531 (*p* < 0.05); One-leg balance (Left), Group *F*_2,51_ = 5.992 (*p* < 0.05), Time *F*_1,51_ = 4.857 (*p* > 0.05), G × T *F*_2,51_ = 10.115 (*p* < 0.05); One-leg balance (Right), Group *F*_2,51_ = 4.818 (*p* < 0.05), Time *F*_1,51_ = 5.919 (*p* < 0.05), G × T *F*_2,51_ = 10.627 (*p* < 0.05); Time Up Go, Group *F*_2,51_ = 5.171 (*p* < 0.05), Time *F*_1,51_ = 6.872 (*p* < 0.05), G × T *F*_2,51_ = 10.112 (*p* < 0.05); Straight walking velocity Group *F*_2,51_ = 4.892 (*p* < 0.05), Time *F*_1,51_ = 5.117 (*p* < 0.05), G × T *F*_2,51_ = 8.991 (*p* < 0.05). *: *p* < 0.05 vs. Pre-test.

**Figure 5 ijerph-18-01515-f005:**
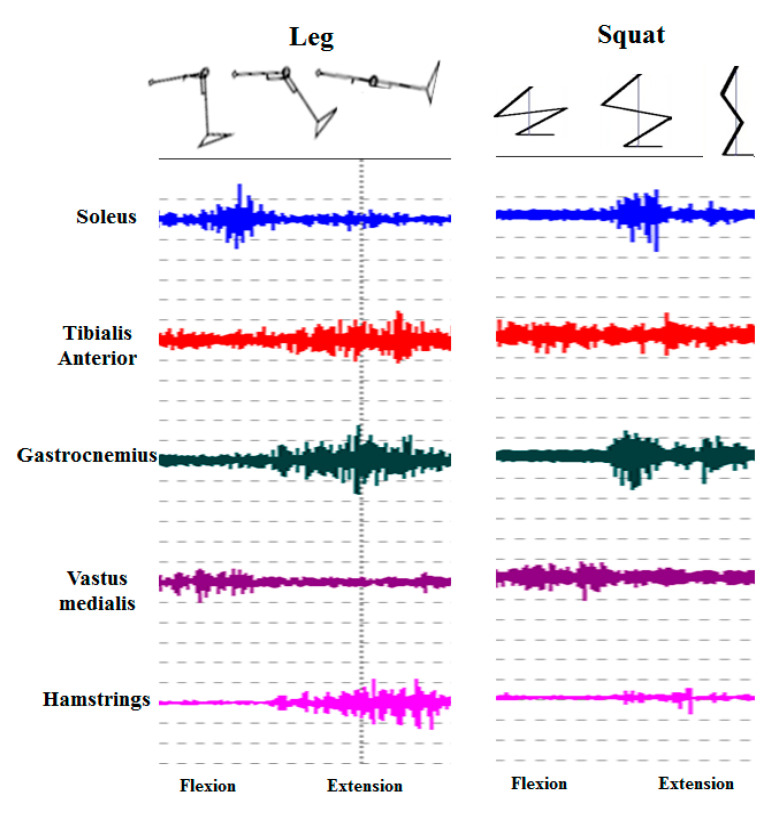
EMG result showing the muscle patterns upon Leg-link system and Squat movements.

**Table 1 ijerph-18-01515-t001:** Combined exercise training program.

Stage	Time (min)	Exercise Program	Set	Frequency	Intensity
Warming-up	5	Lower back Gluteus stretch (single leg)			
		Lower back Gluteus stretch (double leg)			
		Piriformis Gluteus (Medius stretch)			
		Hamstring stretch			
		Quadratus femoris stretch			
Resistance exercise	30	Squat	3	15	60~70%/1RM
		Wide Squat	3	15	
		Tubing band Squat	3	15	
		Tubing band Wide Squat	3	15	
		Single Leg-link exercise	3	15	
		Double Leg-link exercise	3	15	
		Tubing band Leg-link exercise	3	15	
Aerobic exercise	20	Walking & Jogging			60~70%/HRmax
Cool-down	5	Lumbar rotation stretch			
		Lumbar extension Abdominals stretch			
		Cat pose stretch(extension)			
		Cat pose stretch(flexion)			

**Table 2 ijerph-18-01515-t002:** Physical characteristics of subjects.

Group	Age (year)	Height (cm)	Body Weight (kg)	Body Mass Index (kg/m^2^)	Body Fat (%)
Control	73.45 2.17	169.03 4.24	68.57 3.33	24.84 2.71	24.65 3.97
Aerobic Training	73.11 2.11	168.94 1.99	69.55 2.96	25.12 2.11	24.55 2.99
Combined Training	73.19 2.17	168.95 3.95	70.87 4.11	25.11 3.65	23.93 2.81

Values are mean and SD.

**Table 3 ijerph-18-01515-t003:** Changes of temporal parameters of gait after 12 week exercise training.

Item	Control		Aerobic Training		Combined Training		2-Way Repeated ANOVA (F-Value)		
	Pre	Post	Pre	Post	Pre	Post	Group	Time	G × T
Step time (left) (s)	0.54 0.01	0.53 0.01	0.52 0.01	0.52 0.02	0.55 0.01	0.51 * 0.01	2.911	3.561	5.761 (*p* < 0.05)
Step time (right) (s)	0.54 0.01	0.53 0.01	0.56 0.02	0.54 0.02	0.55 0.01	0.51 * 0.01	2.897	3.119	6.717 (*p* < 0.05)
Gait cycle time (left) (s)	1.15 0.05	1.14 0.03	1.44 0.03	1.43 0.04	1.09 0.03	1.02 * 0.03	2.861	4.512	5.019 (*p* < 0.05)
Gait cycle time (right)	1.14 0.06	1.13 0.07	1.12 0.05	1.11 0.03	1.08 0.04	1.01 * 0.02	2.916	3.144	6.719 (*p* < 0.05)
Single support time (left) (%)	34.55 1.36	34.68 1.56	33.44 2.35	33.11 2.11	35.04 1.15	38.17 * 1.21	2.865	3.412	5.910 (*p* < 0.05)
Single support time (right) (%)	35.34 1.01	35.98 1.08	35.44 2.35	36.79 1.20	35.78 1.01	38.81 * 0.96	2.862	3.991	6.123 (*p* < 0.05)
Double support time (left) (%)	28.78 0.95	28.61 1.01	28.76 1.34	27.56 1.23	27.45 0.95	24.54 * 0.01	2.597	3.412	6.002 (*p* < 0.05)
Double support itme (right) (%)	27.99 0.87	28.75 1.01	27.65 1.23	28.75 1.29	27.89 1.17	23.98 * 1.65	2.515	3.412	5.791 (*p* < 0.05)
Ambulation time (s)	3.21 0.33	3.18 0.45	3.11 0.56	3.00 0.23	3.01 0.23	2.54 * 0.37	2.990	3.445	6.100 (*p* < 0.05)
Mean Normalized time	1.05 0.02	1.06 0.03	1.21 0.04	1.29 0.06	1.19 0.02	1.39 * 0.01	2.561	2.991	5.678 (*p* < 0.05)

Values are means and SD, * *p* < 0.05 vs. Pre-test.

**Table 4 ijerph-18-01515-t004:** Changes of spatial parameters of gait after 12 week exercise training.

Item	Control		Aerobic Training		Combined Training		2-Way Repeated ANOVA (F-Value)		
	Pre	Post	Pre	Post	Pre	Post	Group	Time	G × T
Stride width (left) (cm)	48.55 3.21	49.21 3.82	49.87 3.45	50.65 3.12	49.15 3.11	55.62 * 2.95	2.321	2.155	8.512 (*p* < 0.05)
Stride width (right) (cm)	49.44 4.34	49.98 4.01	49.87 3.22	50.12 3.21	48.99 2.82	55.15 * 3.55	2.851	1.456	6.791 (*p* < 0.05)
Stride length (left) (cm)	100.12 5.81	101.45 6.01	100.12 5.89	101.12 5.43	100.32 4.95	110.39 * 6.67	2.442	2.812	6.671 (*p* < 0.05)
Stride length (right) (cm)	101.35 5.76	102.67 5.43	101.23 4.34	103.45 4.12	101.41 5.56	111.34 * 5.90	2.317	3.172	8.231 (*p* < 0.05)
Step/Extremity ratio (left)	0.65 0.02	0.66 0.01	0.65 0.01	0.67 0.01	0.64 0.03	0.74 * 0.02	2.111	3.002	7.099 (*p* < 0.05)
Step/Extremity ratio (right)	0.66 0.03	0.67 0.02	0.66 0.03	0.67 0.03	0.65 0.01	0.72 * 0.08	2.213	2.656	8.919 (*p* < 0.05)
H-H base of support (left)	8.85 0.01	8.31 0.01	8.84 0.03	8.76 0.04	8.34 0.19	7.21 0.25	2.442	3.111	3.101
H-H base of support (right)	8.78 0.01	8.28 0.01	8.75 0.78	8.65 0.89	8.29 0.18	7.15 0.27	2.333	2.561	2.431

Values are means and SD, * *p* < 0.05 vs. Pre-test.

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
