# Peer review of "12-Week Exercise Training of Knee Joint and Squat Movement Improves Gait Ability in Older Women"

_ijerph, 2021, doi:10.3390/ijerph18041515_

Round 1

Reviewer 1 Report

This study investigates the effect of a 12-week of combined or a combined training program on physical performance in older women. Although the topic is interesting given the scarcity of data in the literature, there are several aspects that should be revised.

Abstract: specify what combined training means.

Introduction:

The introduction needs to be clear what the practical question is that you are trying to address. How the answer to this question is important to the field as this is not clear or obvious? How is this study and impactful study and not trivial as this needs more clarity as well. The key issue here is to make sure you set up your approach to the problem. The approach to the problem is essential in determining and describing the rationale for the study. You have not given a basic rationale for the choices made for the variables used in the study.

Methods and results:

For scientific reasons, the sample size for trial needs to be planned carefully. The authors did not perform a power analysis to determine the sample size for the study. Finally, other aspects could be improved, such as the presentation of the two-way ANOVA results in the tables; there are no 95% confidence interval and probability values relative to the effect of the group, time and the interaction between group and time. Furthermore, interactions should also be reported in the figures. Also, an ANOVA 3 (groups) X 2 (time) should be performed.

Discussion:

The discussion section is very descriptive and offers limited comparisons to previous research. Similarly, how do practitioner benefit from that? Again, the discussion section fails to relate the findings to this particular application of interest. Authors are therefore encouraged to make substantial changes throughout to improve the overall quality. In the current form the rationale for the study is not clear, the new value is unclear, and I have difficulties finding specific take home messages for practitioners.

Author Response

Response to Reviewer's Comments

This study investigates the effect of a 12-week of combined or a combined training program on physical performance in older women. Although the topic is interesting given the scarcity of data in the literature, there are several aspects that should be revised.

Point 1 : Abstract: specify what combined training means.

Response 1 : Combined training means resistance exercise and aerobic exercise including walking and light jogging. So, Line 13 : we revised as follow

: and combined training group (n = 18). -> and combined training group with resistance and aerobic exercise (n = 18).

Introduction:

Point 2 : The introduction needs to be clear what the practical question is that you are trying to address. How the answer to this question is important to the field as this is not clear or obvious? How is this study and impactful study and not trivial as this needs more clarity as well. The key issue here is to make sure you set up your approach to the problem. The approach to the problem is essential in determining and describing the rationale for the study. You have not given a basic rationale for the choices made for the variables used in the study.

Response 2 : The following sentences and references related to the importance of improving the walking ability of elderly women were added.

‘The need for exercise programs and equipment that can be used to promote active aging and health of the elderly while aiming for a safe and physical fitness level is being emphasized [10-1]. In particular, the importance of the development of an exercise program that can show a higher clinically positive effect in the field safely in order to improve the walking ability of elderly women is more emphasized.’

Ramalho, F.; Santos-Rocha, R.; Branco, M.; Moniz-Pereira, V.; André, H.;Veloso, A. P.; Carnide, F. Effect of 6-month community-based exercise interventions on gait and functional fitness of an older population: a quasi-experimental study. Clinical Interventions in Aging 2018, 13, 595-606

Methods and results:

Point 3 : For scientific reasons, the sample size for trial needs to be planned carefully. The authors did not perform a power analysis to determine the sample size for the study.

Response 3 : In consideration of the pointed out, the information was completed as follows.

‘In order to calculate the appropriate subjects for 3 groups to secure statistical significance, the G-power 3.1.9.2 program was used to analyze based on the effect size of 0.5 and the power of 0.95. As a result, the total sample size was calculated as 51 subjects. In consideration of the dropout rate of about 20% or more, a total of 62 elderly women aged 70 or above who voluntarily participated in the K University health management program. Those who dropped out due to personal reasons, and those whose test results were not reliable were excluded.’

Point 4 : Finally, other aspects could be improved, such as the presentation of the two-way ANOVA results in the tables; there are no 95% confidence interval and probability values relative to the effect of the group, time and the interaction between group and time. Furthermore, interactions should also be reported in the figures. Also, an ANOVA 3 (groups) X 2 (time) should be performed.

Response 4 : The table has been revised drastically by adding the results of the two-way variance repetition from the viewpoint of enhancing the validity of the research results in consideration of the points noted. Also in the figure, the result of the two-way variance repetition distribution was added in the description.

Figure 1. Change of body composition after 12 week exercise training. 2-way repeated ANOVA for group and time : Body weight, Group F2,51=3.234 (p < 0.05), Time F1,51=5.345 (p < 0.05), G×T F2,51=14.512 (p < 0.05); Body fat, Group F2,51=4.167 (p < 0.05), Time F1,51=3.317 (p < 0.05), G×T F2,51=4.512 (p < 0.05); BMI, Group F2,51=4.177 (p < 0.05), Time F1,51=4.117 (p < 0.05), G×T F2,51=8.912 (p < 0.05); Lean body mass, Group F2,51=4.231 (p < 0.05), Time F1,51=2.356 (p > 0.05), G×T F2,51=5.534 (p < 0.05); WHR, Group F2,51=2.189 (p > 0.05), Time F1,51=1.389 (p > 0.05), G×T F2,51=2.667 (p > 0.05). *: p < 0.05 vs Pre-test.\

Figure 2. Change of physical fitness after 12 week exercise training. 2-way repeated ANOVA for group and time : PEI, Group F2,51=4.131 (p < 0.05), Time F1,51=5.129 (p < 0.05), G×T F2,51=9.167 (p < 0.05); Grip strength (Left), Group F2,51=4.123 (p < 0.05), Time F1,51=5.661 (p < 0.05), G×T F2,51=5.178 (p < 0.05); Grip strength (Right), Group F2,51=3.987 (p < 0.05), Time F1,51=4.890 (p < 0.05), G×T F2,51=6.617 (p < 0.05); Back muscular strength, Group F2,51=4.002 (p < 0.05), Time F1,51=4.919 (p > 0.05), G×T F2,51=9.617 (p < 0.05); Sit-up, Group F2,51=5.977 (p < 0.05), Time F1,51=4.657 (p < 0.05), G×T F2,51=9.111 (p < 0.05); Sit & reach, Group F2,51=6.134 (p < 0.05), Time F1,51=5.919 (p < 0.05), G×T F2,51=8.268 (p < 0.05). *: p < 0.05 vs Pre-test. *: p < 0.05 vs Pre-test.

Figure 3. Change of gait ability-related physical fitness after 12 week exercise training. 2-way repeated ANOVA for group and time : S-style walking velocity, Group F2,51=2.131 (p > 0.05), Time F1,51=4.671 (p < 0.05), G×T F2,51=8.357 (p < 0.05); 8 m walking velocity, Group F2,51=1.911 (p > 0.05), Time F1,51=2.112 (p > 0.05), G×T F2,51=1.196 (p > 0.05); Up & down in chair, Group F2,51=5.551 (p < 0.05), Time F1,51=5.981 (p < 0.05), G×T F2,51=9.531 (p < 0.05); One-leg balance (Left), Group F2,51=5.992 (p < 0.05), Time F1,51=4.857 (p > 0.05), G×T F2,51=10.115 (p < 0.05); One-leg balance (Right), Group F2,51=4.818 (p < 0.05), Time F1,51=5.919 (p < 0.05), G×T F2,51=10.627 (p < 0.05); Time Up Go, Group F2,51=5.171 (p < 0.05), Time F1,51=6.872 (p < 0.05), G×T F2,51=10.112 (p < 0.05); Straight walking velocity Group F2,51=4.892 (p < 0.05), Time F1,51=5.117 (p < 0.05), G×T F2,51=8.991 (p < 0.05). *: p < 0.05 vs Pre-test.

Table 3. Changes of temporal parameters of gait after 12 week exercise training

Item

Control

Aerobic training

Combined training

2-way repeated ANOVA

(F-value)

Pre

Post

Pre

Post

Pre

Post

Group

Time

G×T

Step time (left)

(sec)

0.54

0.01

0.53

0.01

0.52

0.01

0.52

0.02

0.55

0.01

0.51*

0.01

2.911

3.561

5.761

(p < 0.05)

Step time (right) (sec)

0.54

0.01

0.53

0.01

0.56

0.02

0.54

0.02

0.55

0.01

0.51*

0.01

2.897

3.119

6.717

(p < 0.05)

Gait cycle time (left) (sec)

1.15

0.05

1.14

0.03

1.44

0.03

1.43

0.04

1.09

0.03

1.02*

0.03

2.861

4.512

5.019

(p < 0.05)

Gait cycle time (right)

1.14

0.06

1.13

0.07

1.12

0.05

1.11

0.03

1.08

0.04

1.01*

0.02

2.916

3.144

6.719

(p < 0.05)

Single support time (left) (%)

34.55

1.36

34.68

1.56

33.44

2.35

33.11

2.11

35.04

1.15

38.17*

1.21

2.865

3.412

5.910

(p < 0.05)

Single support time (right) (%)

35.34

1.01

35.98

1.08

35.44

2.35

36.79

1.20

35.78

1.01

38.81*

0.96

2.862

3.991

6.123

(p < 0.05)

Double support time (left) (%)

28.78

0.95

28.61

1.01

28.76

1.34

27.56

1.23

27.45

0.95

24.54*

0.01

2.597

3.412

6.002

(p < 0.05)

Double support itme (right) (%)

27.99

0.87

28.75

1.01

27.65

1.23

28.75

1.29

27.89

1.17

23.98*

1.65

2.515

3.412

5.791

(p < 0.05)

Ambulation time (sec)

3.21

0.33

3.18

0.45

3.11

0.56

3.00

0.23

3.01

0.23

2.54*

0.37

2.990

3.445

6.100

(p < 0.05)

Mean Normalized time

1.05

0.02

1.06

0.03

1.21

0.04

1.29

0.06

1.19

0.02

1.39*

0.01

2.561

2.991

5.678

(p < 0.05)

Values are means and SD, *p < 0.05 vs Pre-test

Table 4. Changes of spatial parameters of gait after 12 week exercise training

Item

Control

Aerobic training

Combined training

2-way repeated ANOVA

(F-value)

Pre

Post

Pre

Post

Pre

Post

Group

Time

G×T

Stride width (left)

(cm)

48.55

3.21

49.21

3.82

49.87

3.45

50.65

3.12

49.15

3.11

55.62*

2.95

2.321

2.155

8.512

(p < 0.05)

Stride width (right) (cm)

49.44

4.34

49.98

4.01

49.87

3.22

50.12

3.21

48.99

2.82

55.15*

3.55

2.851

1.456

6.791

(p < 0.05)

Stride length (left) (cm)

100.12

5.81

101.45

6.01

100.12

5.89

101.12

5.43

100.32

4.95

110.39*

6.67

2.442

2.812

6.671

(p < 0.05)

Stride length (right) (cm)

101.35

5.76

102.67

5.43

101.23

4.34

103.45

4.12

101.41

5.56

111.34*

5.90

2.317

3.172

8.231

(p < 0.05)

Step/Extremity ratio (left)

0.65

0.02

0.66

0.01

0.65

0.01

0.67

0.01

0.64

0.03

0.74*

0.02

2.111

3.002

7.099

(p < 0.05)

Step/Extremity ratio (right)

0.66

0.03

0.67

0.02

0.66

0.03

0.67

0.03

0.65

0.01

0.72*

0.08

2.213

2.656

8.919

(p < 0.05)

H-H base of support (left)

8.85

0.01

8.31

0.01

8.84

0.03

8.76

0.04

8.34

0.19

7.21

0.25

2.442

3.111

3.101

H-H base of support (right)

8.78

0.01

8.28

0.01

8.75

0.78

8.65

0.89

8.29

0.18

7.15

0.27

2.333

2.561

2.431

Values are means and SD, *p < 0.05 vs Pre-test

Discussion:

Point 5 : The discussion section is very descriptive and offers limited comparisons to previous research. Similarly, how do practitioner benefit from that? Again, the discussion section fails to relate the findings to this particular application of interest. Authors are therefore encouraged to make substantial changes throughout to improve the overall quality. In the current form the rationale for the study is not clear, the new value is unclear, and I have difficulties finding specific take home messages for practitioners.

Response 5 : In consideration of the point out, the following sentences have been added with reference to previous studies about the applicability of the program developed in this study.

‘The tendency to fall more often during walking is associated with a number of factors [33]. Especially, exercise intervention to improve muscle strength reduce the rate falls by 23%, but resistance training alone may not lead to reductions in falls [34]. Active control of upright balance during walking requires a sensorimotor control loop that collects sensory information about the movement of the body space, detects deviations from the upright posture, and generates appropriate muscle forces to correct these deviations [35]. Therefore, exercise training consisting of knee joint complex exercise with digitally controlled active motion function and squat exercise is thought to be helpful in improving sensorimotor function along with lower limb muscle function in the elderly. In the case of the elderly, the importance of rehabilitation training to improve walking ability is emphasized, so if a feedback function appropriate to the level of muscle strength is used, the possibility of using it in the actual clinical field will increase.’

  1. Osoba, M. Y.; Rao, A. K.; Agrawal, S. K.; Lalwani, A. K. Balance and gait in the elderly: a contemporary review. Laryngosc. Invest. Otolaryngol. 2019, 4, 1-11. doi: 10.1002/lio2.252
  2. Sherrington, C.; Fairhall, N.; Wallbank, G.; Tiedemann, A.; Michaleff, Z.; Howard, K.; Clemson, L.; Hopewell, S.; Lamb, S. E. (2019). Exercise for preventing falls in older people living in the community (Review). Cochr. Database Syst. Rev. 2019, 1, CD012424. doi: 10.1002/14651858
  3. Reimann, H.; Ramadan, R.; Fettrow, T.; Hafer, J. F.; Geyer, H.; Jeka J. J. Interactions between different age-related factors affecting balance control in walking. Front. Sport Act. Living 2020, 2, 94. doi: 10.3389/fspor.2020.00094

Reviewer 2 Report

BRIEF SUMMARY

The was a longitudinal trial (12-week) in which elderly women (n=54) were randomized to control group (n = 18), aerobic training group (n = 18), and combined training group (n = 18), to evaluate physical fitness, gait ability-related physical fitness, and the temporal and spatial parameters of gait. Authors observed significant improvement in all outcomes in the combined training group, but not in other two groups.

BROAD (MAJOR) COMMENTS:

I congratulate authors on their work. Overall, I found the topic timely and clinically important. My main concerns are:

  1. Poor reporting and structure. Please format the paper according to relevant reporting guidelines such as CONSORT or other relevant. Certain information is lacking or should be provided in separate paragraphs to facilitate reading.
  2. Lack of registration in international clinical trial registry.
  3. Lack of justification for the sample size.
  4. Conclusions. You state “This study confirmed….”. I find this overstatement, as you only conducted within-group comparison. Can you conduct additional analysis to determine whether changes in the outcome were significantly different between-groups ? Second, there is no evidence you addressed the issue of statistical bias due to multiple outcomes testing which questions the validity of the results. Finally, it seems you did not think about potential confounding and as such did not adjust your analysis for any covariates, that is, are there any other factors that could potentially explain the results?
  5. English editing. Although English is very good, I suggest authors to put effort in simplifying /shortening sentences where needed to facilitate reading. For example, lines 62-67.

SPECIFIC COMMENTS

TITLE

Could you please simplify the title? This is difficult to understand: “12-Week Exercise Training of Knee Joint Complex Exercise and Squat Movement…”. Also, simply state older women rather than 70-aged women.

ABSTRACT

Lines 13: “physical fitness”: should it be health-related physical fitness?

Please modify conclusion to better reflect what your data shows (see major comment above)

INTRODUCTION

Lines 62-67: too long

Please restructure the information flow in the introduction so that the aims of the study and your hypotheses and shortly summarized in a paragraph at the end of intro.

METHODS

Please start off with the paragraph entitled Study design, and carefully describe. I would suggest to format the paper according to relevant reporting guidelines such as CONSORT or other relevant. Information on study design, exclusion/inclusion criteria, setting and recruitment procedures is currently missing.

Line 81: “The details of the exercise training are as follows.”. Please delete

Table 1: This is a result and as such should be provided in the results section not here.

Table 2: Please format the table to make it smaller.

In the description of each outcomes, please state what was considered as an improvement or deterioration.

Point 2.3. Have you checked normality of data and used appropriate tests depending on it? Please also explain how did you address issue of statistical bias due to multiple outcomes testing.

RESULTS

Please start off with describing study population i.e., move Table 1 here and describe.

Can the authors examine their results further and see if the beneficial effects in the outcomes were in most participants or largely due to large effects in a few?

Also, can you conduct additional analysis to determine between-group differences (not only within-group difference as currently in the paper).

DISCUSSION

I suggest to start off the discussion with a summary paragraph reminding the reader about the study objectives and main results.

Figure 4: this figure does not add much to the study as you did not conduct any statistical or descriptive analysis on muscle activity data. Thus, it seems redundant.

Author Response

Response to Reviewer's Comments

BRIEF SUMMARY

The was a longitudinal trial (12-week) in which elderly women (n=54) were randomized to control group (n = 18), aerobic training group (n = 18), and combined training group (n = 18), to evaluate physical fitness, gait ability-related physical fitness, and the temporal and spatial parameters of gait. Authors observed significant improvement in all outcomes in the combined training group, but not in other two groups.

BROAD (MAJOR) COMMENTS:

I congratulate authors on their work. Overall, I found the topic timely and clinically important. My main concerns are:

Point 1 : Poor reporting and structure. Please format the paper according to relevant reporting guidelines such as CONSORT or other relevant. Certain information is lacking or should be provided in separate paragraphs to facilitate reading.

Response 1 : We have added this sentence for further consideration and adjustment of the paragraph division in consideration of the point out.

‘The need for exercise programs and equipment that can be used to promote active aging and health of the elderly while aiming for a safe and physical fitness level is being emphasized [10-1]. In particular, the importance of the development of an exercise program that can show a higher clinically positive effect in the field safely in order to improve the walking ability of elderly women is more emphasized.’

Point 2 : Lack of registration in international clinical trial registry.

Response 2 : We have added this sentence for further consideration of the point out.

The tendency to fall more often during walking is associated with a number of factors [33]. Especially, exercise intervention to improve muscle strength reduce the rate falls by 23%, but resistance training alone may not lead to reductions in falls [34]. Active control of upright balance during walking requires a sensorimotor control loop that collects sensory information about the movement of the body space, detects deviations from the upright posture, and generates appropriate muscle forces to correct these deviations [35]. Therefore, exercise training consisting of knee joint complex exercise with digitally controlled active motion function and squat exercise is thought to be helpful in improving sensorimotor function along with lower limb muscle function in the elderly. In the case of the elderly, the importance of rehabilitation training to improve walking ability is emphasized, so if a feedback function appropriate to the level of muscle strength is used, the possibility of using it in the actual clinical field will increase.’

  1. Osoba, M. Y.; Rao, A. K.; Agrawal, S. K.; Lalwani, A. K. Balance and gait in the elderly: a contemporary review. Laryngosc. Invest. Otolaryngol. 2019, 4, 1-11. doi: 10.1002/lio2.252
  2. Sherrington, C.; Fairhall, N.; Wallbank, G.; Tiedemann, A.; Michaleff, Z.; Howard, K.; Clemson, L.; Hopewell, S.; Lamb, S. E. (2019). Exercise for preventing falls in older people living in the community (Review). Cochr. Database Syst. Rev. 2019, 1, CD012424. doi: 10.1002/14651858
  3. Reimann, H.; Ramadan, R.; Fettrow, T.; Hafer, J. F.; Geyer, H.; Jeka J. J. Interactions between different age-related factors affecting balance control in walking. Front. Sport Act. Living 2020, 2, 94. doi: 10.3389/fspor.2020.00094

Point 3 : Lack of justification for the sample size.

Response 3 : In consideration of the points noted, it has been modified as follows.

‘In order to calculate the appropriate subjects for 3 groups to secure statistical significance, the G-power 3.1.9.2 program was used to analyze based on the effect size of 0.5 and the power of 0.95. As a result, the total sample size was calculated as 51 subjects. In consideration of the dropout rate of about 20% or more, a total of 62 elderly women aged 70 or above who voluntarily participated in the K University health management program. Those who dropped out due to personal reasons, and those whose test results were not reliable were excluded.’

Point 4 : Conclusions. You state “This study confirmed….”. I find this overstatement, as you only conducted within-group comparison. Can you conduct additional analysis to determine whether changes in the outcome were significantly different between-groups ?

Response 4 : We revised to ‘suggested’

Point 5 : Second, there is no evidence you addressed the issue of statistical bias due to multiple outcomes testing which questions the validity of the results. Finally, it seems you did not think about potential confounding and as such did not adjust your analysis for any covariates, that is, are there any other factors that could potentially explain the results?

Response 5 : In consideration of the points noted, the following contents have been revised and supplemented.

Point 6 : English editing. Although English is very good, I suggest authors to put effort in simplifying /shortening sentences where needed to facilitate reading. For example, lines 62-67.

Response 6 : In consideration of the points noted, the following contents have been revised.

‘Notably, by providing a personal exercise training program with knee joint complex exercise device (leg-link system) with digitally controlled active motion function and squat exercise developed from a complementary viewpoint for activation of lower limb and core muscles [11] to elderly women aged 70 or above, and the study aims to analyze the efficacy of the exercise training program.’

SPECIFIC COMMENTS

TITLE

Point 7 : Could you please simplify the title? This is difficult to understand: “12-Week Exercise Training of Knee Joint Complex Exercise and Squat Movement…”. Also, simply state older women rather than 70-aged women.

Response 7 : We revised to ‘12-Week Exercise Training of Knee Joint and Squat Movement Improves Gait Ability in Older Women in Older Women’

ABSTRACT

Point 8 : Lines 13: “physical fitness”: should it be health-related physical fitness?

Please modify conclusion to better reflect what your data shows (see major comment above)

Response 8 : We revised to ‘Health-related physical fitness,’

INTRODUCTION

Point 9 : Lines 62-67: too long. Please restructure the information flow in the introduction so that the aims of the study and your hypotheses and shortly summarized in a paragraph at the end of intro.

Response 9 : We revised to ‘Notably, by providing a personal exercise training program with knee joint complex exercise device (leg-link system) with digitally controlled active motion function and squat exercise developed from a complementary viewpoint for activation of lower limb and core muscles [11] to elderly women aged 70 or above, and the study aims to analyze the efficacy of the exercise training program.’

METHODS

Point 10 : Please start off with the paragraph entitled Study design, and carefully describe. I would suggest to format the paper according to relevant reporting guidelines such as CONSORT or other relevant. Information on study design, exclusion/inclusion criteria, setting and recruitment procedures is currently missing.

Response 10 : In consideration of the pointed out, the information was completed as follows.

‘In order to calculate the appropriate subjects for 3 groups to secure statistical significance, the G-power 3.1.9.2 program was used to analyze based on the effect size of 0.5 and the power of 0.95. As a result, the total sample size was calculated as 51 subjects. In consideration of the dropout rate of about 20% or more, a total of 62 elderly women aged 70 or above who voluntarily participated in the K University health management program. Those who dropped out due to personal reasons, and those whose test results were not reliable were excluded.’

Point 11 : Line 81: “The details of the exercise training are as follows.”. Please delete

Response 11 : We deleted ‘The details of the exercise training are as follows.’

Point 12 : Table 1: This is a result and as such should be provided in the results section not here.

Response 12 : Please understand that Table 1 presents the basic characteristics of the study subjects before starting exercise treatment. We move to Results.

Point 13 : Table 2: Please format the table to make it smaller.

In the description of each outcomes, please state what was considered as an improvement or deterioration.

Response 13 : It has been modified

Point 14 : Have you checked normality of data and used appropriate tests depending on it? Please also explain how did you address issue of statistical bias due to multiple outcomes testing.

Response 14 : We revise to ‘As post-hoc, paired t-test was carried out for time per group and one-way ANOVA test with Tukey test was carried out for group per time.’

RESULTS

Point 15 : Please start off with describing study population i.e., move Table 1 here and describe.

Response 15 : We move to Results.

Point 16 : Can the authors examine their results further and see if the beneficial effects in the outcomes were in most participants or largely due to large effects in a few?

Also, can you conduct additional analysis to determine between-group differences (not only within-group difference as currently in the paper).

Response 16 : The table has been revised drastically by adding the results of the two-way variance repetition from the viewpoint of enhancing the validity of the research results in consideration of the points noted. Also in the figure, the result of the two-way variance repetition distribution was added in the description.

Figure 1. Change of body composition after 12 week exercise training. 2-way repeated ANOVA for group and time : Body weight, Group F2,51=3.234 (p < 0.05), Time F1,51=5.345 (p < 0.05), G×T F2,51=14.512 (p < 0.05); Body fat, Group F2,51=4.167 (p < 0.05), Time F1,51=3.317 (p < 0.05), G×T F2,51=4.512 (p < 0.05); BMI, Group F2,51=4.177 (p < 0.05), Time F1,51=4.117 (p < 0.05), G×T F2,51=8.912 (p < 0.05); Lean body mass, Group F2,51=4.231 (p < 0.05), Time F1,51=2.356 (p > 0.05), G×T F2,51=5.534 (p < 0.05); WHR, Group F2,51=2.189 (p > 0.05), Time F1,51=1.389 (p > 0.05), G×T F2,51=2.667 (p > 0.05). *: p < 0.05 vs Pre-test.\

Figure 2. Change of physical fitness after 12 week exercise training. 2-way repeated ANOVA for group and time : PEI, Group F2,51=4.131 (p < 0.05), Time F1,51=5.129 (p < 0.05), G×T F2,51=9.167 (p < 0.05); Grip strength (Left), Group F2,51=4.123 (p < 0.05), Time F1,51=5.661 (p < 0.05), G×T F2,51=5.178 (p < 0.05); Grip strength (Right), Group F2,51=3.987 (p < 0.05), Time F1,51=4.890 (p < 0.05), G×T F2,51=6.617 (p < 0.05); Back muscular strength, Group F2,51=4.002 (p < 0.05), Time F1,51=4.919 (p > 0.05), G×T F2,51=9.617 (p < 0.05); Sit-up, Group F2,51=5.977 (p < 0.05), Time F1,51=4.657 (p < 0.05), G×T F2,51=9.111 (p < 0.05); Sit & reach, Group F2,51=6.134 (p < 0.05), Time F1,51=5.919 (p < 0.05), G×T F2,51=8.268 (p < 0.05). *: p < 0.05 vs Pre-test. *: p < 0.05 vs Pre-test.

Figure 3. Change of gait ability-related physical fitness after 12 week exercise training. 2-way repeated ANOVA for group and time : S-style walking velocity, Group F2,51=2.131 (p > 0.05), Time F1,51=4.671 (p < 0.05), G×T F2,51=8.357 (p < 0.05); 8 m walking velocity, Group F2,51=1.911 (p > 0.05), Time F1,51=2.112 (p > 0.05), G×T F2,51=1.196 (p > 0.05); Up & down in chair, Group F2,51=5.551 (p < 0.05), Time F1,51=5.981 (p < 0.05), G×T F2,51=9.531 (p < 0.05); One-leg balance (Left), Group F2,51=5.992 (p < 0.05), Time F1,51=4.857 (p > 0.05), G×T F2,51=10.115 (p < 0.05); One-leg balance (Right), Group F2,51=4.818 (p < 0.05), Time F1,51=5.919 (p < 0.05), G×T F2,51=10.627 (p < 0.05); Time Up Go, Group F2,51=5.171 (p < 0.05), Time F1,51=6.872 (p < 0.05), G×T F2,51=10.112 (p < 0.05); Straight walking velocity Group F2,51=4.892 (p < 0.05), Time F1,51=5.117 (p < 0.05), G×T F2,51=8.991 (p < 0.05). *: p < 0.05 vs Pre-test.

Table 3. Changes of temporal parameters of gait after 12 week exercise training

Item

Control

Aerobic training

Combined training

2-way repeated ANOVA

(F-value)

Pre

Post

Pre

Post

Pre

Post

Group

Time

G×T

Step time (left)

(sec)

0.54

0.01

0.53

0.01

0.52

0.01

0.52

0.02

0.55

0.01

0.51*

0.01

2.911

3.561

5.761

(p < 0.05)

Step time (right) (sec)

0.54

0.01

0.53

0.01

0.56

0.02

0.54

0.02

0.55

0.01

0.51*

0.01

2.897

3.119

6.717

(p < 0.05)

Gait cycle time (left) (sec)

1.15

0.05

1.14

0.03

1.44

0.03

1.43

0.04

1.09

0.03

1.02*

0.03

2.861

4.512

5.019

(p < 0.05)

Gait cycle time (right)

1.14

0.06

1.13

0.07

1.12

0.05

1.11

0.03

1.08

0.04

1.01*

0.02

2.916

3.144

6.719

(p < 0.05)

Single support time (left) (%)

34.55

1.36

34.68

1.56

33.44

2.35

33.11

2.11

35.04

1.15

38.17*

1.21

2.865

3.412

5.910

(p < 0.05)

Single support time (right) (%)

35.34

1.01

35.98

1.08

35.44

2.35

36.79

1.20

35.78

1.01

38.81*

0.96

2.862

3.991

6.123

(p < 0.05)

Double support time (left) (%)

28.78

0.95

28.61

1.01

28.76

1.34

27.56

1.23

27.45

0.95

24.54*

0.01

2.597

3.412

6.002

(p < 0.05)

Double support itme (right) (%)

27.99

0.87

28.75

1.01

27.65

1.23

28.75

1.29

27.89

1.17

23.98*

1.65

2.515

3.412

5.791

(p < 0.05)

Ambulation time (sec)

3.21

0.33

3.18

0.45

3.11

0.56

3.00

0.23

3.01

0.23

2.54*

0.37

2.990

3.445

6.100

(p < 0.05)

Mean Normalized time

1.05

0.02

1.06

0.03

1.21

0.04

1.29

0.06

1.19

0.02

1.39*

0.01

2.561

2.991

5.678

(p < 0.05)

Values are means and SD, *p < 0.05 vs Pre-test

Table 4. Changes of spatial parameters of gait after 12 week exercise training

Item

Control

Aerobic training

Combined training

2-way repeated ANOVA

(F-value)

Pre

Post

Pre

Post

Pre

Post

Group

Time

G×T

Stride width (left)

(cm)

48.55

3.21

49.21

3.82

49.87

3.45

50.65

3.12

49.15

3.11

55.62*

2.95

2.321

2.155

8.512

(p < 0.05)

Stride width (right) (cm)

49.44

4.34

49.98

4.01

49.87

3.22

50.12

3.21

48.99

2.82

55.15*

3.55

2.851

1.456

6.791

(p < 0.05)

Stride length (left) (cm)

100.12

5.81

101.45

6.01

100.12

5.89

101.12

5.43

100.32

4.95

110.39*

6.67

2.442

2.812

6.671

(p < 0.05)

Stride length (right) (cm)

101.35

5.76

102.67

5.43

101.23

4.34

103.45

4.12

101.41

5.56

111.34*

5.90

2.317

3.172

8.231

(p < 0.05)

Step/Extremity ratio (left)

0.65

0.02

0.66

0.01

0.65

0.01

0.67

0.01

0.64

0.03

0.74*

0.02

2.111

3.002

7.099

(p < 0.05)

Step/Extremity ratio (right)

0.66

0.03

0.67

0.02

0.66

0.03

0.67

0.03

0.65

0.01

0.72*

0.08

2.213

2.656

8.919

(p < 0.05)

H-H base of support (left)

8.85

0.01

8.31

0.01

8.84

0.03

8.76

0.04

8.34

0.19

7.21

0.25

2.442

3.111

3.101

H-H base of support (right)

8.78

0.01

8.28

0.01

8.75

0.78

8.65

0.89

8.29

0.18

7.15

0.27

2.333

2.561

2.431

Values are means and SD, *p < 0.05 vs Pre-test

DISCUSSION

Point 17 : I suggest to start off the discussion with a summary paragraph reminding the reader about the study objectives and main results.

Response 17 : The order of the sentences was adjusted according to the pointed out, and the following sentence was placed at the beginning of the discussion.

In this study, after the 12-week exercise training, combined training group showed positive changes in body composition accompanied by improvement in the physical fitness factors, the individual parameters that influence the ambulatory competence, and parameters related to gait ability, especially gait pattern. Thus, the resistance exercise training focusing on knee joint complex exercise device (leg-link system) with digitally controlled active motion function and squat movements for strengthening the lower limb and core muscles, respectively, seems to have had positive influence on physical fitness factors that are crucial in ambulatory competence and on temporal and spatial parameters of gait ability, thereby exerting outstanding effects on improving the ambulatory competence.

Point 18 : Figure 4: this figure does not add much to the study as you did not conduct any statistical or descriptive analysis on muscle activity data. Thus, it seems redundant.

Response 18 : It shows the characteristics of the training equipment applied in this study, and if the importance is not very high, the size is reduced and presented.

Round 2

Reviewer 1 Report

Authors addressed all my comments and suggestions 

Author Response

Response to Reviewer 1-2nd Comments

Authors addressed all my comments and suggestions 

Response : Thank you very much for your kind reviewing.

Reviewer 2 Report

I thank authors for their reply. However, some of my points have been either replied poorly or not at all. 

Point 1. Please format methods section according to relevenat reporting guidelines (CONSORT or STROBE)

Point 4: Please correct in the abstract too.

Point 6. You have not revised the paper and not even the sentence I used as an example. This is too long and difficult to understand. There are many more such long sentences throughout the paper that need to be simplified.

Point 9: as in point 6

Point 10: this is just description of sample size. what about the rest of issues I asked for in the comment?

Point 14. Please state whether you checked normality of data?

Point 17: I do not see that implemenetd in the revised paper

Author Response

Response to Reviewer 2-2nd Comments

Point 1. Please format methods section according to relevenat reporting guidelines (CONSORT or STROBE)

Response 1 : Yes, it has been modified as follows.

2.1. Study Design

In order to analyze the exercise training effect of the subjects, this study measured the items such as body composition, health-related physical fitness, gait ability-related physical fitness, and ambulatory competence, respectively, before and after 12-week exercise training, and compared the change patterns. The participants were divided into control group, aerobic training group, and combined training group. The subjects in aerobic training group and combined training group were asked to participate in the exercise training program for at least three times a week. Control group performed only normal life while being controlled not to perform exercise program for 12 weeks. We obtained approval for the recruitment and exercise treatment experiment from the Institutional Review Board (IRB) of Keimyung University (40525-201802-BR-1218-02).

Point 4: Please correct in the abstract too.

Response 2 : Yes, it has been modified as follows.

This study suggested that the exercise training consisting of knee joint complex exercise with digitally controlled active motion function and squat exercise for strengthening lower extremities and core muscles had positive effects on enhancing the ambulatory competence in elderly women.

Point 6. You have not revised the paper and not even the sentence I used as an example. This is too long and difficult to understand. There are many more such long sentences throughout the paper that need to be simplified.

Response 6 : Overall, I did my best to fix it as following etc.

Thus, based on the recognition that gait function impairment due to sarcopenia in aging is mainly caused by reduced muscular function at the core and lower extremities, this study aims to analyze the level of improved muscular function and variations among factors related to gait ability, by carrying out a combined exercise training for increasing the level of diverse physical fitness factors with special focus on the muscle mass and strength at the core and lower extremities.

Revised to -> Therefore, this study aims to analyze the changes between the level of improvement in muscle function and factors related to gait ability, based on the recognition that the gait dysfunction caused by sarcopenia during aging is mainly due to the decrease in the muscle function of the core and lower extremities.

Point 9: as in point 6

Response 9 : Overall, I did my best to fix it as follow etc.

It is of note that the factors considered to have significant impact on reducing the stride width in the elderly, such as reduced arm swing, reduced joint range of motion in the hip, knee, and ankle, increase in double support time and in H-H base of support [26], were shown to have mostly improved in the group that performed the exercise training focusing on the digitally controlled active motion function knee joint complex exercise and squat movements investigated in this study.

Revised to -> It is considered to be a significant influencing factor in reducing stride in older adults, such as decreased arm swing, decreased range of motion in the hips, knees and ankles, increased double support time, and HH bass [26]. Most of these factors were found to be improved in the group that performed exercise training focused on the digitally controlled active motor function knee joint complex exercise and squat exercise investigated in this study.

Point 10: this is just description of sample size. what about the rest of issues I asked for in the comment?

Response 10 : Yes, it has been modified as follows.

2.1. Study Design

In order to analyze the exercise training effect of the subjects, this study measured the items such as body composition, health-related physical fitness, gait ability-related physical fitness, and ambulatory competence, respectively, before and after 12-week exercise training, and compared the change patterns. The participants were divided into control group, aerobic training group, and combined training group. The subjects in aerobic training group and combined training group were asked to participate in the exercise training program for at least three times a week. Control group performed only normal life while being controlled not to perform exercise program for 12 weeks. We obtained approval for the recruitment and exercise treatment experiment from the Institutional Review Board (IRB) of Keimyung University (40525-201802-BR-1218-02).

2.2. Participants

In order to calculate the appropriate subjects for 3 groups to secure statistical significance, the G-power 3.1.9.2 program was used to analyze based on the effect size of 0.5 and the power of 0.95. As a result, the total sample size was calculated as 51 subjects. In consideration of the dropout rate of about 20% or more, a total of 62 elderly women aged 70 or above who voluntarily participated in the K University health management program. Those who dropped out due to personal reasons, and those whose test results were not reliable were excluded. The participants were divided into control group, aerobic training group, and combined training group, each group containing 18 subjects.

2.3. Exercise Training Program

Point 14. Please state whether you checked normality of data?

Response 14 : It has been further modified as follows.

To analyze the difference between the time before exercise training program and the time after the training, 2-way repeated ANOVA was carried out with the group and period of time as independent variables after first testing the homogeneity of the measurement results.

Point 17: I do not see that implemenetd in the revised paper

Response 17 : Missed by mistake. I added the following in discussion’s first part.

In this study, after the 12-week exercise training, combined training group showed positive changes in body composition accompanied by improvement in the physical fitness factors, the individual parameters that influence the ambulatory competence, and parameters related to gait ability, especially gait pattern. Thus, the resistance exercise training focusing on knee joint complex exercise device (leg-link system) with digitally controlled active motion function and squat movements for strengthening the lower limb and core muscles, respectively, seems to have had positive influence on physical fitness factors that are crucial in ambulatory competence and on temporal and spatial parameters of gait ability, thereby exerting outstanding effects on improving the ambulatory competence.
